# Systematic Evaluation of Modular Robotic Manipulation Policies via Structured Condition Space: A Study on Precision Pick-and-Place Tasks

Anonymous

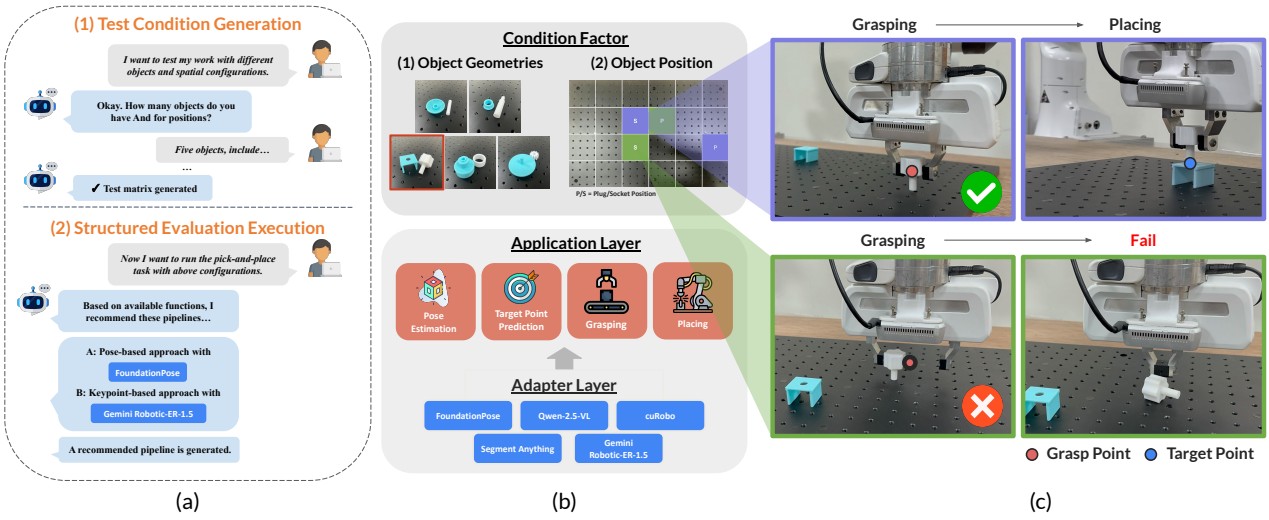

Fig. 1: **Overview of the Proposed Systematic Evaluation Framework for Robotic Manipulation Policies.** (a1) A user interacts with an LLM agent to construct structured, parameterized condition subspaces from high-level evaluation factors. (a2) The agent further compiles robotic manipulation policies within the unified framework. (b) Example condition factors (e.g., object geometry and object position) are shown at the top, while the bottom illustrates the modular architecture comprising an application layer and an adapter layer for policy compilation. (c) The resulting structured condition space enables extensive evaluation and systematic reliability boundary identification. Supplementary video is available at this link[1]

*Abstract*— **Foundation models have demonstrated strong potential for robotic manipulation, promising adaptability across diverse tasks and environments. Despite favorable benchmark performance, these systems often exhibit degraded or unstable behavior under variations in object geometry, spatial configuration, sensing conditions, and workspace constraints—challenges that are amplified in real-world deployment. Existing evaluation efforts broaden evaluation across multiple perturbation axes but primarily focus on condition-level sensitivity. This offers limited insight into the precise configurations under which policies become unreliable. We propose a systematic evaluation framework that explicitly constructs a structured condition space with LLM assistance. Leveraging the semantic and commonsense priors encoded in large language models, we decompose high-level evaluation factors into structured, parameterized subspaces, enabling scalable exploration of environmental variations. This design shifts evaluation from coarse condition-level analysis to structured reliability boundary identification. We further introduce a modular architecture that compiles robotic manipulation policies within this unified framework and supports execution analysis across diverse conditions. Experimental results on precision pick-and-place tasks demonstrate that enables fine-grained characterization of performance degradation and failure patterns, providing actionable insights for robustness assessment and real-world deployment.**

## I. INTRODUCTION

Robustness remains a central challenge for deploying robotic manipulation systems in real-world settings. Despite promising benchmark results, foundation models—including large perception models [1], [2], [3], Vision-Language models (VLMs) [4], [5], [6], [7], and Vision-Language-Action (VLA) models [8], [9], [10], [11], [12]—often exhibit degraded or unstable behavior when exposed to variations in object geometry, spatial configuration, sensing viewpoint, or workspace constraints [13], [14]. Understanding where and how these systems fail under structured real-world conditions is a necessary step toward improving their robustness.

Recent efforts have sought to scale and standardize real-world evaluation of general-purpose policies [15], [16], [17]. However, these works primarily assess condition-level sensitivity—such as object pose variation or semantic rephrasing—without characterizing how performance degrades within each factor. A less explored but equally critical question is under which configurations a policy becomes

---

[1]Video link: `https://tinyurl.com/robot-eval-video`

unreliable, as this finer-grained characterization provides actionable guidance for improving robustness in deployment.

To this end, we propose a systematic evaluation framework that explicitly constructs a structured condition space with LLM assistance, transforming evaluation from coarse condition-level sensitivity analysis to fine-grained reliability boundary identification. Rather than manually enumerating perturbations, we leverage LLMs to decompose high-level condition factors into structured, parameterized subspaces, enabling scalable exploration of policy reliability boundaries.

Our main contributions are as follows:

1) We propose a systematic evaluation framework based on structured condition-space construction, which organizes operating conditions into explicit, factorized test matrices, enabling fine-grained characterization of performance variations across individual condition factors.

2) We design a three-layer modular architecture with interchangeable adapters, enabling controlled comparisons across pipeline configurations under identical operating conditions.

3) Through extensive real-world evaluation on precision pick-and-place tasks, we uncover structured reliability boundaries and provide actionable insights for improving the robustness of manipulation pipelines.

## II. RELATED WORK

### A. Real-World Robot Evaluation

The rise of general-purpose manipulation policies has created a growing demand for scalable and standardized evaluation in real-world settings. RoboArena [15] addresses this by proposing a decentralized protocol that aggregates evaluation results across multiple laboratories, enabling large-scale policy comparison without centralizing experimental infrastructure. AutoEval [16] complements this by introducing an automated platform that standardizes execution and environment reset procedures, reducing the human effort required for repeatable real-world testing. STAR-Gen [17] further advances this direction by introducing a structured taxonomy that organizes environmental variations into visual, semantic, and behavioral axes, providing a principled basis for reasoning about what conditions to test. While these works significantly expand the breadth, scalability, and structure of real-world evaluation, they primarily focus on comparing policy performance across diverse conditions. An equally important but less addressed question is how performance varies systematically along a specific condition factor. Our framework addresses this by explicitly constructing a structured condition space, enabling fine-grained characterization of reliability boundaries within each factor.

### B. High Precision Manipulation

Precision assembly—such as peg insertion, gear meshing, and plug-socket mating—is a fundamental challenge in industrial robotics, requiring accurate perception and tight geometric tolerances. The NIST Assembly Task Board [18] establishes a standardized benchmark for evaluating such tasks. Following this benchmark, a line of work, Factory [19], [20], [21], progressively advances from simulation to real-world robotic execution on NIST-inspired assets. AutoMate [22] further broadens the scope by introducing 100 diverse assembly geometries, enabling the study of precision assembly policies over a significantly wider range of part configurations.

## III. SYSTEMATIC EVALUATION FRAMEWORK

The proposed framework operates across two phases: **(1) condition space construction** and **(2) structured evaluation execution**. In the first phase, the user specifies high-level evaluation goals, and an LLM generates concrete condition instances. In the second phase, a task-specific modular pipeline is composed and executed under each condition, producing stage-level execution signals and task-level outcomes for subsequent analysis.

### A. Modular Architecture and Pipeline Execution

We adopt a three-layer modular architecture that separates decision-making, functional abstraction, and model integration. The Orchestration Layer, powered by a large language model (LLM), dynamically composes a task-specific manipulation pipeline and manages data flow between stages. The Application Layer defines functional modules that can be composed into pipelines, while the Adapter Layer provides standardized wrappers around diverse third-party models, enabling a plug-and-play mechanism for model substitution without modifying pipeline logic. To support systematic evaluation across condition instances, manipulation functions are equipped with built-in checkers that monitor execution status and return a binary pass/fail signal. If a stage fails, the pipeline triggers an early exit. In addition, recorders can be attached to stages to capture relevant data—such as RGB-D images, point clouds, and robot states—after execution. These mechanisms enable precise failure localization and structured data logging across different test conditions.

### B. Structured Condition-Space Construction

Evaluation planning is formulated as the structured construction of a test matrix over explicitly defined condition factors, where each factor represents a dimension of variation that may influence system performance, such as scene configuration, viewpoint, or background. Each factor is associated with a finite set of discrete values or sampling ranges. The Cartesian combination of selected factor values defines the trial configuration space. An LLM assists in transforming factor definitions into concrete condition instances—for example, discretizing workspace position into predefined spatial regions or enumerating object geometry variants—enabling scalable and reproducible exploration of configuration boundaries.

For each trial, the physical environment is configured according to the corresponding condition values and reset between trials. The composed manipulation pipeline is then executed autonomously, with stage-level checkers monitoring

execution status. Execution outcomes are automatically associated with the corresponding condition in the test matrix, ensuring traceable logging per condition.

## IV. CASE STUDY: MODULAR PIPELINES FOR PRECISION PICK-AND-PLACE

We conduct a case study on precision plug-and-socket assembly to validate the proposed framework on a real-world manipulation task requiring accurate perception under varying object geometries and spatial configurations. The subject of evaluation in this study is not an end-to-end learned manipulation policy, but two modular, training-free pipelines for precision assembly that share identical orchestration logic and manipulation functions while differing only in their perception component, enabling controlled comparison of perception robustness within a consistent execution backbone.

### A. Task Definition

The precision pick-and-place task requires the robot to grasp a plug and transport it to an insertion-ready pose above the corresponding socket, achieving accurate spatial alignment without physical insertion. At the start of each trial, both objects are placed at random positions in a stable, upright orientation on the workspace surface. Successful execution requires reliable localization of both objects, stable grasp execution, and accurate placement. The experimental setup is described in **Appendix A**.

### B. Pipeline Configurations

We adopt two pipeline configurations that share identical orchestration logic, manipulation functions, and execution sequence (plug localization, socket localization, grasping, and placing), differing only in their perception adapters: a pose-based approach (Pipeline A) using SAM2 [23] and Foundation Pose [3] for coarse-to-fine 6D pose estimation, and a keypoint-based approach (Pipeline B) using Gemini Robotics-ER 1.5 [7] for direct 2D keypoint prediction. Details are provided in the **Appendix B**.

### C. Test Matrix Construction

The test matrix defines the experimental conditions under which the two pipelines are evaluated, organized along two primary condition factors: **object geometry** (difference in plug-socket form factors) and **spatial configuration** (placement of the plug and socket within the tabletop workspace). For object geometry, five pairs of plug-socket assemblies are selected from the AutoMate dataset [22] to represent diverse geometric characteristics, differing in overall size, symmetry, surface structure, and grasp affordances (Fig. 2).

For spatial configuration, the tabletop workspace is discretized into a grid and partitioned into three distance-based zones (D1–D3) according to Euclidean distance from the board center (Fig. S5). The resulting test matrix spans 5 object geometries and 20 spatial position pairs per object, yielding 100 unique condition instances, enabling systematic characterization of robustness across diverse operating conditions.

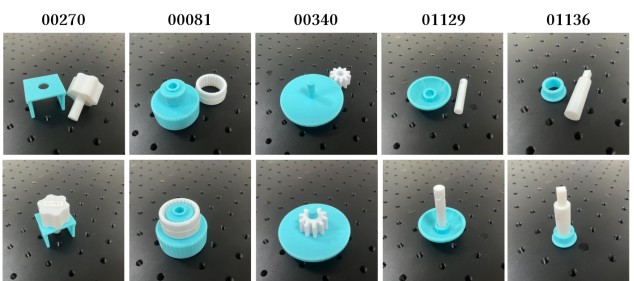

Fig. 2: 3D-printed plug-socket assemblies selected from the AutoMate dataset [22] in this study.

## V. EVALUATION RESULTS

### A. Metrics Definition

We define four metrics to evaluate pipeline performance. Since the plug–socket interfaces are rotationally symmetric, all metrics are defined in terms of 3D positional deviation. **Plug Position Error** and **Socket Position Error** measure the 3D Euclidean distance (mm) between the predicted and ground-truth positions, quantifying perception accuracy at the localization stage. **Grasp Success Rate** is the proportion of trials in which a stable grasp is achieved, determined by gripper width after closure. **Final Execution Error** measures the 3D Euclidean distance (mm) between the plug's final placed position and the target pose, annotated within the same coordinate frame, eliminating the cross-sensor bias. Human annotators review the recorded results after trial completion. Errors exceeding 100mm are excluded as outliers, and final execution error is computed only for successful grasps (Pipeline A: 459/500; Pipeline B: 232/500).

### B. Per-Object Results (Geometry Effects)

Across all five objects, Pipeline A achieves consistently lower mean positional errors than Pipeline B (e.g., plug position error: 12.33 mm vs. 23.13 mm) and a substantially lower Final Execution Error (7.66 mm vs. 17.29 mm), with significantly smaller standard deviations indicating greater stability (see **Appendix C** for detailed results). Object geometry directly influences failure modes: thin and slender objects (#01129, #01136) leave little margin for localization error, resulting in low grasp success rates in Pipeline B (36% and 37%), while flatter objects (#00081, #00340) increase the probability of displacement during grasping even under accurate localization in Pipeline A. These geometry-specific failure patterns demonstrate that structured condition-space evaluation can reveal reliability boundaries inaccessible to condition-level analysis.

### C. Per-Position Results (Spatial Effect)

Pipeline A achieves uniformly high grasp success rates across most spatial configurations, with the exception of the rightmost column (X=5), where success rates drop to 64% despite no corresponding increase in plug position error, suggesting that robot kinematic constraints rather than perception errors are the primary failure source. This insight

TABLE I: Success Rate Across Object IDs

| Object ID | #00081 | #00271 | #00340 | #01129 | #01136 | Overall |
|---|---|---|---|---|---|---|
| Pipeline A | 82% | 99% | 81% | 99% | 98% | 91.8% |
| Pipeline B | 51% | 62% | 46% | 36% | 37% | 45.3% |

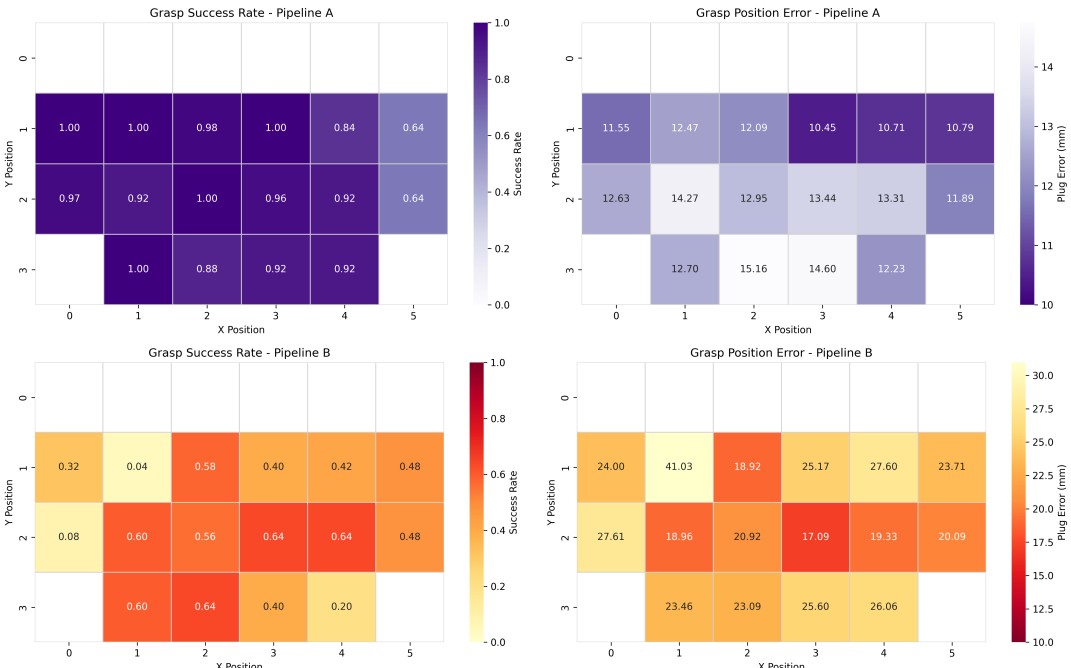

Fig. 3: Spatial performance heatmaps for Pipeline A (top) and B (bottom), showing grasp success rate (left) and the plug position error (right) across the 4x6 grid.

is only discoverable by jointly analyzing perception error and task success across spatial conditions: when perception error is acceptable but success rate remains poor, kinematic constraints should be investigated first.

For Pipeline B, regions with lower position error consistently correspond to higher grasp success rates, indicating that perception accuracy directly drives task success. Spatial performance heatmaps for both pipelines are shown in Fig. 3. The comparison across distance categories (D1–D3) in Figure S6 shows that Pipeline A maintains consistently high accuracy across all distance levels, while Pipeline B exhibits progressive degradation as distance increases, reflecting the limitation of its single-stage perception strategy.

### D. Observations for Real-world Deployment

Based on experiments across two pipelines and five object types, we highlight the following observations to inform real-world deployment: (1) **Pipeline selection by geometry:** Pipeline A is recommended for tasks requiring high localization accuracy; Pipeline B exhibits larger localization error and may not be suitable for precision-critical tasks involving thin or slender objects. (2) **Spatial zone restriction:** Pipeline B may benefit from being restricted to near-to-mid distance zones (D1–D2), or augmented with an active approach mechanism to compensate for distance-induced accuracy loss. (3) **Failure diagnosis:** When perception error is acceptable but

grasp success rate remains low, kinematic constraints may be a primary failure source worth investigating. (4) **Resource trade-offs:** Pipeline A requires local GPU resources but offers higher precision and spatial robustness; Pipeline B operates via cloud-based API with no local GPU requirement, offering a lighter deployment path at the cost of reduced accuracy.

## VI. CONCLUSION

We present a systematic evaluation framework based on structured condition-space construction, enabling fine-grained characterization of where and how manipulation pipelines become unreliable across individual condition factors, and allowing practitioners to identify reliability boundaries and derive actionable improvement directions. In future work, extending LLM involvement beyond condition-space construction to the analysis stage would enable automated interpretation of evaluation results and generation of deployment guidelines, completing the evaluation loop. Combined with the extensibility of the layered architecture, we hope this work can serve as a foundation for systematic robustness evaluation across a wider range of real-world manipulation tasks.

## A. Experimental Setup

Our experimental platform consists of a 7-DoF Franka Emika Panda robot with the default gripper, a wrist-mounted RealSense D435 RGB-D camera for conducting experiments, and a fixed Orbbec Femto Bolt RGB-D camera for recording. 3D-printed assemblies from the AutoMate dataset [22] are placed on a 400 × 600 mm optical breadboard serving as the tabletop workspace, as shown in Fig. S4. To enable systematic spatial variation, the workspace is partitioned into three distance-based zones according to radial distance from the board center. These predefined regions provide a structured basis for sampling plug and socket positions during subsequent test matrix construction.

## B. Pipeline Configurations

Both pipelines share identical orchestration logic and execution sequence: plug localization, socket localization, grasping, and placing. The grasp pose is derived from a canonical grasp defined in the AutoMate [22] dataset. The only difference lies in the perception adapters used to localize the plug and socket.

**Pipeline A: Pose-based Approach.** Extending the perception pipeline introduced in AutoMate [22], this pipeline formulates object localization as full 6D pose estimation using a coarse-to-fine strategy: Qwen2.5-VL [6] provides coarse semantic localization, followed by SAM2 [23] segmentation and FoundationPose [3] for 6D pose estimation. The estimated poses are converted into grasp and placement targets for motion planning.

**Pipeline B: Keypoint-based Approach.** Inspired by VLM-based keypoint prediction applications in robotic manipulation [24], [25], [26], this pipeline formulates perception as manipulation-oriented keypoint localization. Gemini Robotics-ER 1.5 [7] directly predicts 2D keypoints from an RGB image, which are unprojected into 3D coordinates using depth and refined along the Z axis using the CAD model and estimated table height. The resulting 3D keypoints are converted into grasp and placement targets.

TABLE II: Comparison of the two perception pipelines.

|  | Pose-based | Keypoint-based |
|---|---|---|
| Localization Output | 6D pose (SE(3)) | 3D keypoint |
| Estimation Process | Two-stage (coarse-to-fine) | Single-stage |
| Semantic Segmentation | Required | Not required |
| CAD Model Usage | For pose estimation | For Z refinement |

## C. Detailed Evaluation Results

Tables III–V provide detailed per-object quantitative results across all metrics for both pipelines. Note that systematic cross-sensor bias may be introduced in Plug and Socket Position Errors, as predicted positions are obtained from the wrist-mounted RealSense D435 while ground-truth is annotated using the Orbbec Femto; this does not affect cross-pipeline comparison since both pipelines share identical camera configurations.

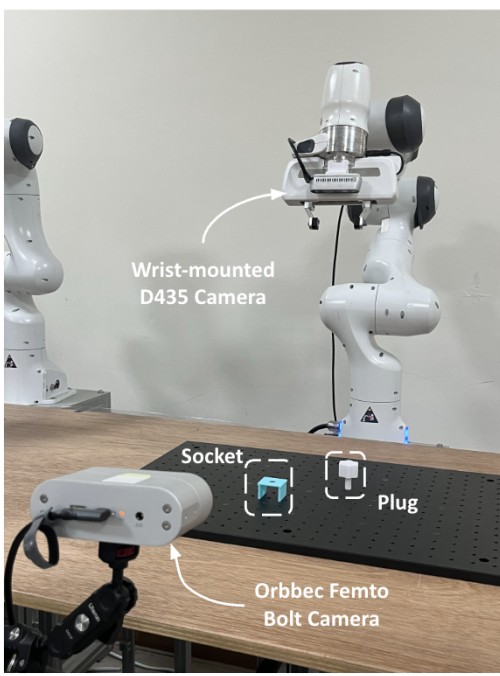

Fig. S4: **Experimental setup.** A Franka Emika Panda robot with a wrist-mounted RealSense D435 camera and an Orbbec Femto Bolt camera is used for recording.

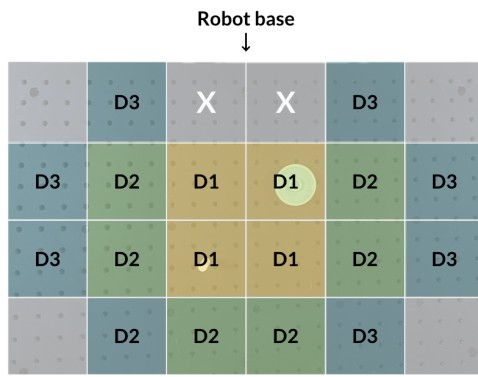

Fig. S5: Tabletop workspace partitioned into three distance-based zones (D1–D3). Cells marked with crosses (X) are excluded due to proximity to the robot base.

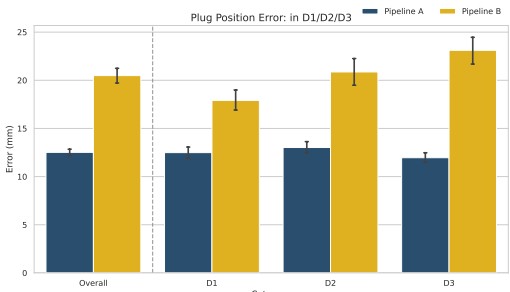

Fig. S6: Comparison of Plug Position Error Across D1–D3 for Pipeline A and B.

TABLE III: Plug Position Error Across Object IDs

| Object ID | #00081 | | #00271 | | #00340 | | #01129 | | #01136 | | Overall | |
|---|---|---|---|---|---|---|---|---|---|---|---|---|
| | Avg | Std | Avg | Std | Avg | Std | Avg | Std | Avg | Std | Avg | Std |
| Pipeline A | 10.95 | 2.10 | 17.75 | 3.05 | 11.21 | 2.78 | 12.00 | 1.96 | 9.72 | 2.12 | 12.33 | 3.72 |
| Pipeline B | 23.46 | 16.35 | 21.92 | 12.34 | 23.66 | 13.61 | 23.78 | 14.50 | 22.84 | 12.38 | 23.13 | 13.94 |

TABLE IV: Socket Position Error Across Object IDs

| Object ID | #00081 | | #00271 | | #00340 | | #01129 | | #01136 | | Overall | |
|---|---|---|---|---|---|---|---|---|---|---|---|---|
| | Avg | Std | Avg | Std | Avg | Std | Avg | Std | Avg | Std | Avg | Std |
| Pipeline A | 17.39 | 3.88 | 17.38 | 4.02 | 17.32 | 4.76 | 14.60 | 3.32 | 13.05 | 3.38 | 15.95 | 4.30 |
| Pipeline B | 18.80 | 8.23 | 20.03 | 11.45 | 22.77 | 11.53 | 18.09 | 9.45 | 19.49 | 8.92 | 19.82 | 10.12 |

TABLE V: Final Position Error Across Object IDs

| Object ID | #00081 | | #00271 | | #00340 | | #01129 | | #01136 | | Overall | |
|---|---|---|---|---|---|---|---|---|---|---|---|---|
| | Avg | Std | Avg | Std | Avg | Std | Avg | Std | Avg | Std | Avg | Std |
| Pipeline A | 7.89 | 2.74 | 6.21 | 2.65 | 7.51 | 5.94 | 10.61 | 3.84 | 6.06 | 2.82 | 7.66 | 4.1 |
| Pipeline B | 18.7 | 8.93 | 17.46 | 13.04 | 19.34 | 11.69 | 16.42 | 15.22 | 13.34 | 9.8 | 17.29 | 12.05 |

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
