# OpenReview forum: "Systematic Evaluation of Modular Robotic Manipulation Policies via Structured Condition Space: A Study on Precision Pick-and-Place Tasks"
_IEEE.org/ICRA/2026/Workshop/Manipulation_Robustness — ICRA 2026_

### Official Review · Reviewer_CiJL · 2026-05-04
**Structured Condition-Space Evaluation for Diagnosing Manipulation Robustness on Precision Plug-socket Pick-and-place task**

**Rating:** 7
**Confidence:** 4

**Review:**

This paper proposes a framework for systematically evaluating modular robotic manipulation pipelines by constructing a structured condition space over factors such as object geometry and object position. Instead of reporting only aggregate success rates, the proposed framework organizes evaluation conditions into a test matrix and analyzes how performance varies across individual factors. The authors demonstrate the framework on a precision plug-socket pick-and-place task, comparing a pose-based pipeline using SAM2 and FoundationPose with a keypoint-based pipeline using Gemini Robotics-ER 1.5. The results show that the pose-based pipeline achieves substantially higher success and lower localization error, while the structured heatmap analysis reveals different failure mechanisms, including perception-driven failures for the keypoint-based pipeline and possible kinematic/ workspace limitations for the pose-based pipeline.

## Pros

- **Timely and relevant problem.** The paper addresses an important issue in real-world robot evaluation: aggregate success rates are often insufficient for understanding when and why manipulation systems fail. The focus on identifying reliability boundaries is well motivated.

- **Clear structured evaluation idea.** The condition-space formulation is intuitive and useful. Organizing evaluation around object geometry and spatial configuration provides more diagnostic information than reporting only a single overall success rate.

- **Practical real-world experiments.** The paper includes real robot experiments over multiple object geometries and spatial configurations. The $5 \times 20$ condition matrix provides a reasonably systematic case study for a workshop paper.

- **Useful failure diagnosis.** The spatial heatmaps are one of the strongest parts of the paper. They show that Pipeline A maintains relatively low localization error but loses success in some workspace regions, suggesting kinematic constraints, whereas Pipeline B’s success appears more directly tied to localization error.

- **Modular architecture is useful.** Separating application-layer functions from adapter-layer models is a practical design choice and makes controlled comparison between perception modules easier.

## Cons

- **The comparison between Pipeline A and Pipeline B is not well balanced.** While Pipeline B provides a useful lightweight and VLM-based comparison point, the experimental results suggest that it is not a practically competitive solution for the precision pick-and-place task studied here. Its low overall success rate and large localization errors make the comparison predictable, since the task naturally favors a 6D pose-estimation pipeline over a single-stage VLM keypoint-based pipeline. This weakens the strength of the experimental claim. It will be great to add an additional task where Pipeline B has a more plausible advantage, such as coarse manipulation, semantic target selection, affordance-based grasping, or settings where primitive shapes or accurate 6D pose estimation are unavailable.

- **Some evaluation details could be clarified.** The paper reports several useful metrics, including plug/ socket localization errors, grasp success rate, and final execution error. However, the overall task success criterion is not explicitly defined, and it is unclear whether success requires both a stable grasp and final placement error below a threshold. Additional details on ground-truth annotation, outlier exclusion, trial randomization, and statistical testing would make the experimental conclusions more rigorous.

- **The role of the LLM is not sufficiently validated.** The paper claims LLM assistance for condition-space construction and pipeline recommendation, but it does not convincingly show why an LLM is necessary or better than a manually designed test matrix. Since the condition factors in the case study are relatively simple, the LLM component currently feels more like an interface layer than a necessary technical contribution.

- **The experimental scope is limited.** The case study focuses on a precision pick-and-place task without physical insertion. This is useful, but it does not fully demonstrate the generality of the proposed framework for broader manipulation tasks, contact-rich manipulation, long-horizon tasks, or learned policy evaluation.

- **The framing of “manipulation policies” is somewhat broader than the experiments.** The evaluated systems are modular, training-free pipelines rather than learned manipulation policies in the usual sense. The title and claims should be adjusted or clarified to avoid overclaiming.

Overall, this is a solid workshop paper that presents a practical and timely framework for diagnosing reliability boundaries in modular robotic manipulation pipelines through structured condition-space evaluation. Its main strength is that it moves beyond aggregate success rates and uses object-geometry and spatial-position analysis to reveal meaningful failure patterns. However, the paper would be stronger with a clearer justification of the LLM component, a more precise distinction between pipeline and policy evaluation, a broader experimental scope, and a more balanced comparison where the keypoint-based VLM pipeline has a plausible advantage. I would recommend acceptance for a workshop, as the work provides a useful evaluation perspective and aligns well with the workshop’s focus on manipulation robustness by helping identify where, when, and why manipulation pipelines become unreliable under structured real-world variations.

---

### Official Review · Reviewer_2NH1 · 2026-05-10
**Review: Systematic Evaluation of Modular Robotic Manipulation Policies via Structured Condition Space**

**Rating:** 6
**Confidence:** 4

**Review:**

## Summary

The paper introduces an evaluation framework for robotic manipulation that uses an LLM to help construct a structured condition space, essentially a test matrix over factors like object geometry and spatial placement. A three-layer modular architecture lets the authors swap out perception components while keeping everything else fixed. They demonstrate this on a precision pick-and-place task, comparing a pose-based pipeline (SAM2 + FoundationPose) against a keypoint-based one (Gemini Robotics-ER 1.5) across 5 objects and 20 spatial configurations, for 100 conditions total.

## Strengths

1. **Well-motivated problem.** The gap between coarse condition-level sensitivity analysis and fine-grained reliability boundary identification is real and practically important. Understanding *where* a policy breaks, not just *that* it breaks, is valuable for deployment.

2. **Actionable findings from spatial analysis.** The observation that Pipeline A's grasp failures at X=5 are driven by kinematic constraints rather than perception errors (since plug position error does not increase there) is a good example of the kind of insight this framework can produce. Similarly, Pipeline B's progressive degradation across distance zones D1→D3 is clearly shown.

3. **Practical deployment recommendations.** Section V-D distills concrete guidance (geometry-based pipeline selection, spatial zone restriction, failure diagnosis heuristics, resource trade-offs) that practitioners could act on.

## Weaknesses

1. **The experimental scope is too narrow to back up the generality claims.** One task, two pipelines, two condition factors, five objects. The abstract and introduction talk about sensing viewpoint, workspace constraints, background variation — none of these appear in the experiments. The paper is also motivated by the brittleness of VLA and foundation model policies, yet neither pipeline is a learned end-to-end policy. Calling this a "systematic evaluation framework" sets expectations the case study doesn't meet. It reads more like a solid single-task analysis than a general framework contribution.

2. **Statistical rigor is lacking.** The spatial heatmap in Fig. 3 is one of the paper's key results, but it's unclear how many trials each cell represents. With 100 total trials per pipeline spread over a 4×6 grid (and some cells excluded), individual cells might have as few as 5 trials. Values like 0.04 and 0.08 in Pipeline B's heatmap could easily be one success out of 25 or one out of 12, the difference between these is not meaningful without confidence intervals. No significance tests are reported anywhere. For a paper about systematic evaluation, this is a notable gap.

3. **No baseline evaluation framework for comparison.** The paper positions itself against RoboArena, AutoEval, and STAR-Gen, but does not compare the insights produced by this framework against what those methods would reveal on the same task. Without such a comparison, the claim that structured condition-space evaluation reveals "reliability boundaries inaccessible to condition-level analysis" is asserted rather than demonstrated.

## Questions

1. How many trials per cell in Fig. 3? At that sample size, are the observed differences statistically meaningful?

2. How sensitive is the generated test matrix to prompt phrasing? Could you provide the actual prompts used?

---

### Decision · Program_Chairs · 2026-05-21

Accept